# Synthesis of Silver Oxide Nanoparticles: A Novel Approach for Antimicrobial Properties and Biomedical Performance, Featuring *Nodularia haraviana* from the Cholistan Desert

**DOI:** 10.3390/microorganisms11102544

**Published:** 2023-10-12

**Authors:** Lubna Anjum Minhas, Muhammad Kaleem, Amber Jabeen, Nabi Ullah, Hafiz Muhammad Umer Farooqi, Asif Kamal, Farooq Inam, Abdulwahed Fahad Alrefaei, Mikhlid H. Almutairi, Abdul Samad Mumtaz

**Affiliations:** 1Department of Plant Sciences, Faculty of Biological Sciences, Quaid-i-Azam University, Islamabad 45320, Pakistan; lubnaminhas@bs.qau.edu.pk (L.A.M.); mkaleem@bs.qau.edu.pk (M.K.); amberjabeen58@gmail.com (A.J.); farooqinam2015@gmail.com (F.I.); 2Board of Governors Regenerative Medicine Institute, Cedars-Sinai Medical Center, West Hollywood, Los Angeles, CA 90048, USA; 3Department of Zoology, College of Science, King Saud University, P.O. Box 2455, Riyadh 11451, Saudi Arabiamalmutari@ksu.edu.sa (M.H.A.)

**Keywords:** *Nodularia haraviana*, Cholistan, silver nanoparticles, antifungal, antibacterial, antileishmanial, biocompatibility, cytotoxicity

## Abstract

Nanoparticles have emerged as a prominent area of research in recent times, and silver nanoparticles (AgNPs) synthesized via phyco-technology have gained significant attention due to their potential therapeutic applications. *Nodularia haraviana*, a unique and lesser-explored cyanobacterial strain, holds substantial promise as a novel candidate for synthesizing nanoparticles. This noticeable research gap underscores the novelty and untapped potential of *Nodularia haraviana* in applied nanotechnology. A range of analytical techniques, including UV-vis spectral analysis, dynamic light scattering spectroscopy, scanning electron microscopy, Fourier transform infrared spectroscopy, and X-ray powder diffraction, were used to investigate and characterize the AgNPs. Successful synthesis of AgNPs was confirmed through UV-visible spectroscopy, which showed a surface plasmon resonance peak at 428 nm. The crystalline size of AgNPs was 24.1 nm. Dynamic light scattering analysis revealed that silver oxide nanoparticles had 179.3 nm diameters and a negative surface charge of −18 mV. Comprehensive in vitro pharmacogenetic properties revealed that AgNPs have significant therapeutic potential. The antimicrobial properties of AgNPs were evaluated by determining the minimum inhibitory concentration against various microbial strains. Dose-dependent cytotoxicity assays were performed on Leishmanial promastigotes (IC_50_: 18.71 μgmL^−1^), amastigotes (IC50: 38.6 μgmL^−1^), and brine shrimps (IC_50_: 134.1 μg mL^−1^) using various concentrations of AgNPs. The findings of this study revealed that AgNPs had significant antioxidant results (DPPH: 57.5%, TRP: 55.4%, TAC: 61%) and enzyme inhibition potential against protein kinase (ZOI: 17.11 mm) and alpha-amylase (25.3%). Furthermore, biocompatibility tests were performed against macrophages (IC_50_: >395 μg mL^−1^) and human RBCs (IC_50_: 2124 μg mL^−1^). This study showed that phyco-synthesized AgNPs were less toxic and could be used in multiple biological applications, including drug design and in the pharmaceutical and biomedical industries. This study offers valuable insights and paves the way for further advancements in AgNPs research.

## 1. Introduction

Nanoparticles are widely studied due to their diverse applications across biology, catalysis, optics, pharmaceutics, health, agriculture, and the industrial sector [1,2,3]. Various physical and chemical procedures are utilized to produce these nanoparticles, which can be costly and require high temperatures and pressures. The chemicals involved can have negative environmental and health impacts on researchers [4,5]. In recent times, there has been growing interest in eco-friendly synthesis methods for nanoparticle production, mainly through plant-based biogenic synthesis, which has proven to be a quick, easy, and efficient process that eliminates the need for hazardous chemicals typically utilized in organic synthesis [6,7,8]. AgNPs have been synthesized from various macroalgae and microalgae. They have been used for several applications, such as in cytotoxicity, antioxidant, antimicrobial, and anticancer agents [9]. Researchers find the development of silver nanoparticles (AgNPs) highly appealing due to the metal’s nobility and extensive utility in various fields, particularly in biomedical and biochemistry applications. The synthesis of AgNPs by algal cells is believed to be facilitated by specific metabolites, such as enzymes, that reduce silver ions into AgNPs [10].

Plants, fungi, actinomycetes, algae, diatoms, bacteria, lichen, and cyanobacteria have been reported to be efficient in reducing metal precursors to their corresponding nanoparticles, making them an attractive option for phyco-synthesis of metal nanoparticles [11]. It has been described in previous studies that nanoparticles have been synthesized from plants using their extract as a reducing and capping agent [12,13,14]. Cyanobacteria, in particular, are a sustainable resource with significant biotechnological applications due to their rich bioactive compound repository and their presence in freshwater, marine, and terrestrial ecosystems [15]. Additionally, some cyanobacteria species can fix atmospheric nitrogen and sequester metal ions from aquatic environments through biosorption or bioaccumulation, making them an ideal platform for metal-based nanoparticle synthesis [16].

Cyanobacteria also contain various biomolecules, such as proteins, lipids, carbohydrates, pigments, and secondary metabolites, that offer antimicrobial and anticancer properties [17]. Many cyanobacterial strains have been utilized for the synthesis of metallic NPs like silver (Ag), gold (Au), and platinum (Pt), which act either extracellularly or intracellularly. For instance, *Oscillatoria willei* NTDM01 [18], *Oscillatoria limnetica* [19], *Desertifilum* sp. [17], *Nostoc linckia* [20], *Nostoc* sp. [7], and *Leptolyngbya* sp. L-2 [8] have been used for the biogenic synthesis of AgNPs. Cyanobacteria are an ancient and widely distributed group of photoautotrophic prokaryotes in diverse ecosystems, including freshwater, marine, and terrestrial habitats. In addition to their ability to perform oxygenic photosynthesis, numerous cyanobacterial species possess a distinctive capability to convert atmospheric nitrogen (N_2_) into ammonia by reducing nitrogen gas. Nitrogen reductase enzymes facilitate this conversion [15].

This research explores the phyco-synthesis of AgNPs utilizing the newly discovered and characterized *Nodularia haraviana* as an eco-friendly, cost-effective, abundant, and renewable source with improved performance. The metabolites in the aqueous extract of the cyanobacterial strain act as oxidizing and reducing agents and capping agents. This is the first study to report the phyco-synthesis of AgNPs using *Nodularia haraviana* extract. The green synthesis of AgNPs was subjected to various analytical techniques such as XRD, UV, FTIR, DSL, SEM, and EDX to gain further insights into their properties. Moreover, multiple bioactivities were carried out, including antibacterial, antifungal, cytotoxicity, biocompatibility, alpha-amylase, and protein kinase inhibition assays. By thoroughly examining the synthesized nanoparticles and assessing their effectiveness, we ensured their suitability for diverse applications while prioritizing safety and minimizing potential harm.

## 2. Materials and Methods

### 2.1. Chemicals

Various chemicals and catalysts used for this research were of scientific grade and obtained from different chemical suppliers.

### 2.2. Nodularia haraviana and Their Environmental Conditions

*Nodularia haraviana*, a cyanobacterium, was successfully isolated from the natural ponds in Pakistan’s Cholistan desert in Punjab province. To establish a pure culture of *Nodularia haraviana*, we created an axenic inoculum and introduced it into 500 mL Erlenmeyer flasks containing BG 11 medium [15]. The stationary growth phase was then obtained after 22 days of incubation at 28 °C and continuous illumination at an intensity of 57.75 Mmol m^−2^ s^−1^. Once the colony’s purity had been established, it was kept for further analysis. A light microscope was utilized to examine the morphological features of the cyanobacterium, including its form, color, length, and breadth. The strain cells were collected through a centrifugation (HERMLE Labortechnik GmbH, Wehingen, Germany) process for 15 min at 10,000 rpm. The particles were then washed with sterile water after the supernatant was removed. Glass plates were used to dry the cyanobacterial biomass for 1 day at 55% relative humidity. The dried powder was utilized in this experiment without supplying nutrients. To prepare an aqueous extract of *Nodularia haraviana*, 5 g of cyanobacterial fine powder was mixed with 500 mL of deionized water and heated for 1 h at 100 °C. The mixture was centrifuged at 3000 rpm for 20 min; the clear brown solution was separated from the large particles at the bottom of the conical tube and filtered using filter paper.

### 2.3. Phyco-Synthesis of AgNPs

The extract was then mixed with a 1 mM AgNP solution in a ratio of 1:9 (10 mL extract to 90 mL AgNP solution) (Figure 1). The mixture was heated on a hot plate (SCILOGEX MS7-H550-S) for 2 h at 70 °C with continuous stirring, aiming for an accelerated reaction rate. The pH of the resulting solution was determined to be 9.73. A significant color transformation from pale yellow to brown was observed, indicating the reduction of Ag+ to Ag0 [19]. The resulting pellet was washed multiple times with distilled water to eliminate any remaining particles. Following this, it was incubated for 2 h at 75 °C. The dried powder material was finely ground using a pestle and mortar, and any impurities were eliminated through calcination in an open-air furnace at 400 °C for 2 h. The calcined AgNPs were stored in a dry, cool, and dark environment for further characterization. 

### 2.4. Structural and Morphological Characterization of AgNPs 

Various analytical methods, including chemical, physical, and morphological investigations, were used to describe the AgNPs. UV-vis spectroscopy (UV-4000 spectrophotometer, Berlin, Germany) was utilized to measure the phyco-synthesis of AgNPs between 800–300 nm. FTIR (Thermo scientific Fisher Nicolete IS10, Waltham, MA, USA) with a resolution of 1 cm^−1^ was used to investigate the biomolecules present in the fresh biomass of *Nodularia haraviana* and contribute to the bioreduction of AgNPs. The structural properties and crystal purity of the AgNPs were demonstrated using the X-ray powder diffraction (XRD) (PANalytical, Almelo, The Netherlands, Europe) method, which examined 2 theta values between 10 and 70. A scanning electron microscope (JSM5910 JEOL, Tokyo, Japan) fitted with an energy-dispersive X-ray spectroscopy detector was used to investigate the surface morphology of phyco-synthesized AgNPs. Energy-dispersive X-ray spectroscopy (Malvern Zetasizer Nano, Tokyo, Japan) analysis was utilized to examine the elemental composition of AgNPs. Using a DLS technique, the dispersion and average particle size of AgNPs were studied.

### 2.5. Biological Potential of AgNPs

Different biological activities were conducted to investigate the biological potential of the synthesized nanoparticles, following relevant guidelines and regulations. 

#### 2.5.1. Analysis of Biocompatibility Potential

The ability to cause cell lysis and the level of toxicity exhibited by the synthesized AgNPs were evaluated using the method previously described by Minhas et al. [8]. The red blood corpuscles pellet was obtained from a human blood sample by centrifuging at 226× *g* for 10 min and purified using PBS at pH 7.4. This experimental procedure involved the addition of aliquots of red blood cells to a solution containing silver nanoparticles (AgNPs) at various concentrations in the range of 300 to 6.25 μg mL^−1^. The total volume was made up to 1 mL using PBS, and the solution was then incubated at room temperature for 60 min to allow any reactions or interactions between the components. The supernatant was centrifuged. The absorbance of the sample was measured at 540 nm. Hemolysis was verified using Triton-X 100 (TTX-100) as a positive control, while a blood sample mixed with PBS was utilized as a negative control. Each experiment was performed in triplicate.

The biocompatible nature of the phyco-synthesized AgNPs was also evaluated using HMs by the previously described method [21]. The HMs were grown in RPMI media that contained antibiotics (pen-strep), 10% fetal bovine serum (FBS), and 25 mM herpes. The HMs were then plated in 96-well dishes and incubated with CO_2_ (5%) for 1 day. The HMs were subjected to various concentrations of NPs ranging from 300 to 6.2 μg L^−1^ for hours to evaluate the effects of the phyco-synthesized AgNPs. The amount of hemoglobin was measured using a microplate reader at 540 nm wavelength. Triton-X 100 was used as a reference standard for the current study.

#### 2.5.2. Assessment of Cytotoxicity Potential of Bio-AgNPs

The cytotoxicity potential of phyco-synthesized AgNPs was evaluated using the brine shrimp (BS) lethality test [22]. *Artemia salina* (AS) eggs were placed in a sterile saltwater container with a volume of 1000 mL. Continuous aeration was provided for 48 h; after hatching, 10 larvae were collected and transferred to glass vials, and each vial was filled with 4.5 mL of saltwater. Subsequently, 0.5 mL of AgNPs at several concentrations (ranging from 6.25 to 300 μg mL^−1^) was added to each vial of nauplii. The vials were then exposed to light conditions for 24 h at room temperature. Following the treatment, the number of surviving nauplii in each vial was counted, and the percentage of mortality was calculated. GraphPad software 8.0 was used to calculate the IC_50_ values. 

*Leishmania tropica* was used to study the antileishmanial potential of silver nanoparticles (AgNPs), including both the promastigote and amastigote forms of the parasite. To provide *Leishmania tropica* cells with the nutrients they needed to proliferate, 10% FBS was added to the MI99 medium during *Leishmania tropica* culture. AgNPs were generated in a 96-well plate with varying concentrations (from 6.25 to 300 μg mL^−1^) and incubated (5% CO_2_) for 72 h at 24 °C to evaluate their impact. This controlled environment allowed the *Leishmania tropica* cells to be exposed to the AgNPs and for any potential effects to occur. DMSO was used as a negative control, while amphotericin B was a positive control. After treatment, surviving *Leishmania tropica* were counted, and the absorbance at 540 nm was determined. GraphPad software was used to calculate the IC_50_ values.

#### 2.5.3. Analysis of Antioxidant Activities

The antioxidant capacity of AgNPs was investigated using the spectrophotometric method [23,24]. The solution was made by mixing 25 mL of methanol with 2.4 mg of the free radical 2,2-diphenyl-1-picrylhydrazyl (DPPH). The AgNPs were treated at various concentrations (300 to 6.25 μg mL^−1^). In this experiment, ascorbic acid was the positive control, and DMSO was the negative control. A total volume of 200 µL comprising 20 µL of AgNPs and 180 µL of the reagent solution was utilized for making the reaction mixture. Then, these mixes were kept in complete darkness for 2 h. The spectrophotometer was used to measure the absorbance of the sample at 517 nm wavelength [25]. The total antioxidant capacity (TAC) was measured using the phosphomolybdenum method [26]. Results were represented as micrograms of ascorbic acid equivalents per milligram of the material examined. The sample absorbance was measured at 695 nm wavelength. DMSO was utilized as a negative control, while ascorbic acid was a positive control. The total reducing power (TRP) was measured using the potassium ferricyanide technique [27]. The positive control was ascorbic acid, while the negative control was DMSO. The absorbance of the sample was measured at a wavelength of 630 nm, and milligrams of ascorbic acid equivalents were used to assess the TRP.

#### 2.5.4. Anti-Diabetic Assay 

The alpha-amylase inhibition activity assay was conducted to evaluate the anti-diabetic activity of AgNPs [28]. The assay involved incubating amylase (0.5 mg mL^−1^) without and with the extract and the standard drug Acarbose at 25 °C for 10 min. It was created as a buffer solution of sodium phosphate with a pH of 6.9 and a concentration of 0.02 M. Different amounts of the produced AgNPs, ranging from 28.125 to 900 μg mL^−1^, were added to this buffer solution. Then, the solution was incubated for 10 min at 25 °C. The buffer solution was the to C1% starch and agitated for 30 min. The progress of the enzymatic reaction was stopped by adding dinitrosalicylic acid to the buffer solution. The solution was then placed in a water bath at 100 °C for 15 min. After that, the solution was allowed to cool to room temperature, and the sample absorbance was measured at 540 nm wavelength. A combination of solutions devoid of nanoparticles (NPs) was utilized as the negative control, while a solution containing Acarbose was used as the positive control.

#### 2.5.5. Protein Kinase Inhibition Assay

The following procedures were utilized to investigate the responsiveness of Streptomyces 85E strains, which use kinase for hyphae growth and formation. The bacterial strains (BS) were cultured in tryptic soy broth and refreshed for 24 to 48 h. The culture was swabbed onto mineral ISP4 media after being refreshed. The filter paper discs were carefully positioned on marked plates after being infused with a sample solution (5 L of 20 mg mL^−1^ DMSO). Surfactant- and DMSO-infused discs were utilized as positive controls, respectively. The plates were then kept at 28 °C for 48–72 h to encourage the growth of hyphae. The clear or bald growth inhibition zone surrounding the discs was quantified using a vernier caliper. Clear or barren regions around the discs denote phosphorylation inhibition, which limits mycelium development and spores. Bald areas show the sample’s propensity to obstruct mycelium growth, indicating an inhibitory impact. Blank areas, conversely, signify cytotoxicity and the toxic effects of the test items [29,30]. 

#### 2.5.6. Evaluation of Antibacterial and Antifungal Assays

Using the agar disc diffusion method, the antibacterial activity of AgNPs was examined against harmful bacteria such as *Staphylococcus aureus*, *Klebsiella pneumoniae*, *Pseudomonas aeruginosa*, and *coagulase-negative Staphylococcus* [7,31,32]. After being incubated in nutrient broth for 18 h at 37 °C, bacterial cultures were transferred to the nutrient agar medium, and wells were made using a cork borer. Then, each well received 50 µL of synthetic silver nanoparticles, and the plates underwent an 18 h incubation period at 37 °C. To investigate the antibacterial impact of the AgNPs, the clear zone of inhibition around each well was measured and computed [33]. The food poisoning method assessed AgNPs’ antifungal efficacy [7]. On PDA medium, preserved fungal strains were cultivated for 72 h at 26.1 °C. Silver oxide nanoparticles were added to potato dextrose agar medium in various quantities, from 50 to 200 μg mL^−1^. A 4 mm diameter fungal inoculum disc was gently put into the center of each PDA plate using a cork borer. PDA plates without nanoparticles were utilized as positive controls to compare the outcomes. The diagrammatic representations of AgNPs are shown in Figure 2.

### 2.6. Statistical Analysis

We evaluated the AgNPs synthesized using phyco-material for antibacterial, cytotoxicity, enzyme inhibition, DPPH radical scavenging, total antioxidant activity, and biocompatibility experiments. According to three replicates for each sample, the findings are presented as means and standard deviations. Microsoft Excel 365, GraphPad Prism (8.0), Orinig Lab 2023 (10.0), and other software programs were used for the statistical analysis.

## 3. Results 

The cyanobacterial strain was characterized using light microscopy, which showed the morphology and purity of the strain. The filamentous structure of the selected cyanobacterial isolate was observed to be straight or curved. Based on its morphological characteristics, the cyanobacterial strain was identified as *Nodularia haraviana* (Figure 3a). The chemical reduction process was used to synthesize AgNPs. The color change of the solution into yellowish-brown was observed, indicating the successful synthesis of AgNPs through the phyco-synthesis process (Figure 3b). The yellowish color of the solution showed the formation of colloidal nanoparticles dominated by AgNPs. Hence, adding *Nodularia haraviana* extracts resulted in a color change [34].

UV-visible spectroscopy is an important technique used to verify the synthesis and stability of metal nanoparticles in an aqueous solution. When AgNPs were made from algal powder and distributed in aqueous media, the UV-visible spectra showed a unique (SPR) surface plasmon resonance in the 300–800 nm wavelength range. It was noted that after 3 h of the reaction, the silver nanoparticles made with *Nodularia haraviana* powder showed an SPR band with a wavelength of 428 nm [35]. This band’s intensity gradually increased as a function of the reaction period (Figure 4a) [36]. 

FTIR analysis was used to investigate the biofunctional groups responsible for Ag ion (Ag+) bioreduction into AgNPs (Ag0) and the capping of the synthesized silver oxide nanoparticles. Many peaks were observed, as depicted in Figure 4b and tabulated in Table 1. The band at 3526 cm^−1^ was ascribed to the stretching vibrations of the O-H bonds found in alcohols or phenols [22], and the band between 2921 cm^−1^ (C–H stretch) to the alkane group [37]. Additionally, the peaks at 2015 cm⁻¹ and 1598 cm⁻¹ corresponded to the stretching vibrations of the N=C=S (Isothiocyanate) and N-H (amine) functional groups, respectively. The peak at 1097 cm⁻¹ and 845 cm⁻¹ corresponded to the stretching vibrations of the C-O and C-Cl bonds, signifying the existence of alcohols and chlorides or chloroalkanes. Finally, the vibrational band depicted at the 517 cm^−1^ wavelength affirmed the presence of silver oxide (Ag-O) and agreed with the published literature [38,39].

The crystal quality of the AgNPs synthesized through phyco-synthesis was assessed using XRD spectrum analysis, as shown in Figure 4c. The chromatic spectrum exhibited eight prominent Bragg’s reflection peaks at 2θ degrees, with corresponding values of 31.85°, 34.45°, 36.34°, 37.41°, 45.88°, 47.64°, 56.70°, and 69.18°. These peaks corresponded to the (110), (111), (002), (101), (200), (220), (311), and (201) groups of matrix planes of AgNPs, indicating a face-centered cubic structure. This observation aligns closely with the JCPDS index (Card No. 00-024-0072), thus confirming the crystalline nature of AgNPs in a cubic face-centered hexagonal structure. The mean crystallite size of the AgNPs was determined to be 24.1 nm (Figure 4d). The phyco-synthesized AgNPs were found to be biphasic, with slight changes in the peak position, indicating the presence of particles in a crystalline form. Therefore, the XRD chromatic diagram confirmed that the AgNPs synthesized through the green method possessed a crystalline nature. This finding is similar to studies by Rajasekar et al. [40] and Anjali et al. [41], which demonstrated that aqueous extracts could phyco-synthesize silver nanoparticles with nanocrystalline properties.

To investigate the surface properties of AgNPs, an SEM image was taken, as presented in Figure 5a. The presence of white-colored dots in the image indicated the successful synthesis of AgNPs, and these particles exhibited a spherical morphology. It was observed that the capping reagent used to stabilize NPs did not interact with the collected aggregates, which confirmed that only the extracts played a crucial role in the reduction process [42]. Moreover, some larger AgNP aggregates formed due to the agglomeration of smaller particles, as evident in the SEM image.

The presence of silver in the suspension was confirmed through EDX analysis. Eight peaks between 1 kV and 9 kV were observed in the presented spectrum. Specifically, AgNPs exhibited a distinct absorption peak between 0.5 and 1 kV (Figure 5b). The quantitative analysis further confirmed a high content of silver (65.38%) in the sample synthesized by *Nodularia haraviana*. However, the analysis also revealed the presence of other elements, such as O, S, C, and Na, with percentages of 23.63%, 1.27%, 1.84%, and 2.14%, respectively. It is possible that these elements were present due to some salt or protein residue in the cyanobacterial strain isolates from deserted area [37]. 

Dynamic light scattering was used to determine the thickness of the capping or stabilizing compound enveloping metallic particles, as well as the average size distribution of the AgNPs in the solution, which was found to be 179.3 nm, and the polydispersity index (PDI), which was found to be 1.00 (Figure 5c and Table 2). The zeta potential value was determined to evaluate the synthesized AgNPs’ surface charge and measure the charge amount. This value supported the stability of the particles in dispersion by demonstrating the existence of certain charged groups on the nanoparticles’ surface [43]. The findings (Figure 5d and Table 2) revealed that the AgNPs synthesized by *Nodularia haraviana* extract exhibited a negative charge of −18 mV, which may have been due to the adsorption of free nitrate ions present in the solution, providing a repulsive force as electrostatic stabilization [43,44]. Salvioni et al. [45], published research exhibiting a mainly negative charge of AgNPs fabricated with citric and tannic acid.

### 3.1. Biological Application

#### 3.1.1. Biocompatibility with Human RBCs and Macrophages (HMs)

As demand for the clinical application of silver nanoparticles (AgNPs) continues to increase, it is crucial to ensure their biosafety. Therefore, it is essential to investigate the hemolytic potential of AgNPs, particularly for medical devices that come into contact with blood. It is crucial to consider the physicochemical features of AgNPs and their biocompatibility estimation for the materials used in these devices. Hemolysis is a condition that occurs when the membrane of erythrocytes is compromised, leading to the leakage of hemoglobin, which can result in harmful health conditions [46].

Using biogenic AgNPs, the biocompatibility of these nanoparticles was investigated concerning the integrity of human erythrocytes. The level of hemoglobin release was assessed using spectrophotometry, and the results indicated that an increase in AgNP concentration led to induced hemolytic action on the erythrocytes. The data revealed that the percentage of cell lysis was 0.2%, 0.76%, 2.6%, 3.2%, 5.3%, 7.1%, 9.4%, 11.3%, and 14.7%, corresponding to 6.5–300 μg mL^−1^ AgNP concentrations, respectively (Figure 6a). Triton X-100 positive control and PBS buffer negative control exhibited 78.64% and 0.011% hemolysis, respectively. Although the mode of action of AgNPs in inducing hemolysis is still unclear, it is believed that metallic silver undergoes ionization upon contact with bodily fluids, releasing Ag+ in response to particle surface area [47]. In biological systems, the rapid binding of Ag+ to anionic ligands like chloride (Cl-), inorganic sulfide, and thiols (-SH) can lead to a decrease in the availability of Ag+ for cell destruction [48]. The toxicity of nanosilver to red blood cells (RBCs) may be attributed to the release of free silver ions, the overall concentration of silver ions, and/or the interaction between cellular components and nanoparticles [48]. 

To confirm the biocompatibility of the synthesized AgNPs, human macrophages (HMs) were also utilized in this study. To initiate the experiment, the HMs were seeded into a 96-well plate and cultured in RPMI medium for 24 h to ensure cell attachment. The cells were then exposed to varying concentrations of silver oxide NPs ranging from 6.25 μg mL^−1^ to 300 μg mL^−1^. A dose-dependent response of the HMs to the AgNPs was observed, with the cells being inhibited by 25.48% at the highest concentration of 300 μg mL^−1^, suggesting the biocompatibility of the AgNPs [7]. Studies have shown that macrophages have inherent mechanisms to inhibit the detrimental effects of reactive oxygen species (ROS) generated from external sources. Lower levels of ROS production do not exhibit toxicity to macrophages or red blood cells as long as the concentration remains below a certain threshold [49]. It has been reported that an increase in the concentration of ROS can lead to toxicity in RBCs and macrophages [49]. The calculated IC_50_ value of the AgNPs was determined to be >395 μg mL^−1^, further supporting the biocompatibility of the synthesized AgNPs. Figure 6b displays the results of the biocompatibility assay performed on the AgNPs. 

#### 3.1.2. Anti-Diabetic Activity

Diabetes mellitus is a prevalent disease worldwide. Insulin deficiency causes an increase in blood glucose levels, leading to diabetes mellitus. Anti-diabetic drugs are used to artificially regulate blood glucose levels. Several studies have been conducted to produce an effective drug against diabetes. The present study offers a promising drug for improving antidiabetic activity. The α-amylase enzymatic inhibition technique evaluated the anti-diabetic activity of the *Nodularia haraviana* extract and silver oxide nanoparticles. The percentage inhibition of *Nodularia haraviana* extract, AgNPs, and the standard reagent increased with increasing concentration, as shown in the graph (Figure 7a) [28]. The percentage inhibition of AgNPs was tested at various concentrations ranging from 28.125, 56.25, 112.5, 225, and 450, to 900 μg mL^−1^, with excellent inhibition observed. The maximum anti-diabetic activity of AgNPs was 25.3% at a concentration of 500 μg mL^−1^. Saratale et al. [50] reported similar results for the α-amylase activity of AgNPs using acarbose as the standard drug. These findings indicate the potential of AgNPs to scavenge different concentration levels and provide anti-diabetic activity.

#### 3.1.3. Protein Kinase Inhibition Activity

Phosphorylation of tyrosine and threonine/serine protein kinases plays a vital role in regulating various biological processes, including proliferation of cells, apoptosis, cell differentiation, and metabolism. These metabolic reactions are regulated during tumorigenesis, driven by genetic alterations that contribute to cancer development. Consequently, protein kinase has emerged as a potential target for inhibiting critical cancer pathways. The action of protein kinases is crucial for *Streptomyces* sp. aerial hyphae formation. Recent studies have demonstrated that silver nanoparticles (AgNPs) can function as protein kinase inhibitors, potentially impeding the growth of *Streptomyces* sp. aerial hyphae [51]. Visual indicators such as clear or bald areas surrounding a disc were employed to assess the inhibition of phosphorylation and the subsequent production of spores and mycelium. The AgNPs showed efficacy in a dose-dependent manner, with a maximum zone of inhibition of 17.11 mm and 14.44 mm at 900 and 450 μg mL^−1^, respectively (Figure 7b) [30]. Hence, the results suggest that AgNPs were more effective inhibitors when tested against *Streptomyces* strains.

## 4. Cytotoxicity Assay

The brine shrimp bioassay lethality method has successfully assessed bioactive compounds’ cytotoxicity and pharmacological effects [52] and the mechanism of action is shown in Figure 8. This technique was used in our work to evaluate the cytotoxicity of AgNPs made from phyco-material. As shown in Figure 9a and Table 3, it was investigated how varying AgNPs concentrations (from 6.25 to 300 μg mL^−1^) affected the mortality of *A. salina* brine shrimp. The findings showed a clear relationship between rising AgNP concentrations and a considerable increase in the death rate of *A. salina* brine shrimp. After 24 h, AgNPs at a concentration of 6.25 μg mL^−1^ showed a 2.1% inhibition, but doses of 150–300 μg mL^−1^ showed 28%, 37%, and 39% inhibition. AgNPs were shown to have an IC_50_ value of 134.1 μg mL^−1^, indicating a comparatively low toxicity level to *A. salina*. The biosynthesized AgNPs were less harmful to brine shrimp [22]. These findings reveal a new use for desert-derived AgNPs, indicating that they have high cytotoxic potential. The mechanism of action of AgNP cytotoxicity is shown in Figure 8. The same result was reported in some previous studies [53,54].

Leishmaniasis, caused by protozoan parasites of the genus Leishmania, is a widespread parasitic disease found in more than 100 countries globally [55]. The existing drugs used to treat this chronic disease often exhibit limited effectiveness, high costs, and potential toxicity. Despite the formulation of antimonials as potential therapeutic agents for leishmaniasis, the parasites have developed resistance to these drugs. As a result, scientists are looking for other medications to treat leishmaniasis. To assess the anti-leishmanial capability of the AgNPs, we exposed leishmanial promastigotes and amastigotes to a range of dosages (6.25–300 μg mL^−1^) for 72 h. Table 3 and Figure 9 shows that AgNPs had promising potential, with IC_50_ values of 18.71 μg mL^−1^ for promastigotes and 38.6 μg mL^−1^ for amastigotes [33]. Additionally, it was discovered that the promastigotes were more vulnerable than the amastigotes. These NPs may be employed in targeted drug delivery to treat leishmanial parasites, according to the reduced IC_50_ values. We believe that our work is the first to document the antileishmanial efficacy of phyco-synthesized AgNPs employing *Nodularia haraviana.* The same result was reported in some previous studies [56,57].

### 4.1. Alpha-Amylase Macrophages

#### 4.1.1. Antioxidant Activity

Antioxidant-rich natural substances have become more significant in the fight against illness and cellular damage. Interest in these substances has increased the demand for antioxidants in medical and pharmaceutical applications. AgNPs made using *Nodularia haraviana* are shown to have antioxidant action in Figure 10 and Table 4. Using a DPPH free radical scavenging activity (FRSA) experiment, it was possible to determine if antioxidant species known as radical scavengers were present on the surfaces of AgNPs. According to earlier research on biosynthesized AgNPs [28], the assay’s results showed an FRSA potential of 57.51%. With an IC50 value of 62.12 μg mL^−1^ in our work, the aqueous solution of silver oxide nanoparticles demonstrated considerable dose-dependent DPPH radical scavenging action [13,42].

In the TAC, an antioxidant converted molybdate (VI) to molybdate (V) in an acidic environment, resulting in a molecule with a green hue that could be detected by spectrophotometry at a wavelength of 695 nm [53]. With 61% inhibition at a concentration of 300 μg mL^−1^, the current investigation showed that an aqueous solution of AgNPs displayed a dose-dependent solid response. In the reducing power test, the antioxidant reduced Fe (III) to Fe (II) by transferring an electron. This reaction created a bluish-green hue in the mixture that could be measured spectrophotometrically at a wavelength of 700 nm [58]. The aqueous solution of silver oxide nanoparticles showed considerable dose-dependent reducing power activity, according to a recent work by [8]. It was discovered that the inhibition percentage was 55.4 μg mL^−1^.

#### 4.1.2. Antimicrobial Assays 

The disc-diffusion method (Figure 11), was used to evaluate the antibacterial activity of AgNPs at concentrations ranging from 50 to 200 μg mL^−1^ against *Bacillus subtilis, Pseudomonas aeruginosa, Escherichia coli, Coagulase-negative Staphylococcus, Staphylococcus aureus, and Klebsiella pneumonia* (Figure 12a and Table 5). Most bacterial strains were found to be susceptible to AgNPs, with coagulase-negative *Staphylococcus* and *Staphylococcus aureus* being the most susceptible strains (MIC: 50 μg mL^−1^), and *Bacillus subtilis* being the least susceptible strain (MIC: 150 μg mL^−1^). The positive control used was Oxytetracycline (10 μg). Not a single concentration of AgNPs was more effective than the positive control. Overall, AgNPs showed potential bactericidal activity against different bacterial strains due to the adsorbed functional groups on the surface of the nanoparticles, resulting in a dose-dependent effect. Previous results have shown that the antibacterial potential of AgNPs is due to ROS generation, membrane damage, and surface defects in the symmetry of NPs (Figure 11) [59]. Moreover, the bioactive functional moieties from *Nodularia haraviana* extracts that resulted in capped AgNPs played an essential role in antibacterial activity.

Biogenic AgNPs have been reported to have limited antifungal potential, while extensive research studies have been conducted on their antibacterial potential. Table 6 and Figure 12b illustrate the effectiveness of AgNPs in controlling the growth of pathogenic fungi. The study employed the food poisoning method and tested the 50–200 μg mL^−1^ concentration range against *Alternaria alternata* and *Botrytis cinerea.* Most of the tested fungal strains were found to be susceptible to AgNPs. *Botrytis cinerea* showed the highest susceptibility, with a minimum inhibitory concentration (MIC) of 50 μg mL^−1^, while *Alternaria alternata* showed the least susceptibility, with a MIC of 100 μg mL^−1^ [59]. Previous research studies suggest that the interaction of AgNPs with fungal hyphae, mycelia, and spores inhibits fungal growth and ROS generation. Overall, AgNPs have demonstrated a significant dose-dependent effect, and none of the concentrations used showed more potential than the positive control. These results are similar to those of earlier studies on the antimicrobial properties of *Nodularia haraviana*-mediated AgNPs [12,60]. A comparison of *Nodularia haraviana*-mediated AgNPs with previous studies is described in Table 7.

## 5. Conclusions

This is the first study to establish a simple, cost-effective, and eco-friendly process for synthesizing AgNPs using Nodularia haraviana extract. Furthermore, AgNPs were characterized using several spectroscopic and microscopic techniques to elucidate the stability and functionality of AgNPs. Additionally, AgNPs were investigated for various biomedical properties, including antibacterial, antifungal, and antileishmanial activity, biocompatibility assays, and antioxidant activities. However, it was determined that the therapeutic potential of AgNPs varied with different concentrations. In the future, considering the evolution of disease resistance to currently available drugs, biogenic AgNPs are expected to become potential antimicrobial and cytotoxic agents alone or in combination with other FDA-approved drugs. Further studies are encouraged on the mechanistic and synthesis aspects of the AgNPs by applying various cyanobacterial aqueous extracts.

## Figures and Tables

**Figure 1 microorganisms-11-02544-f001:**
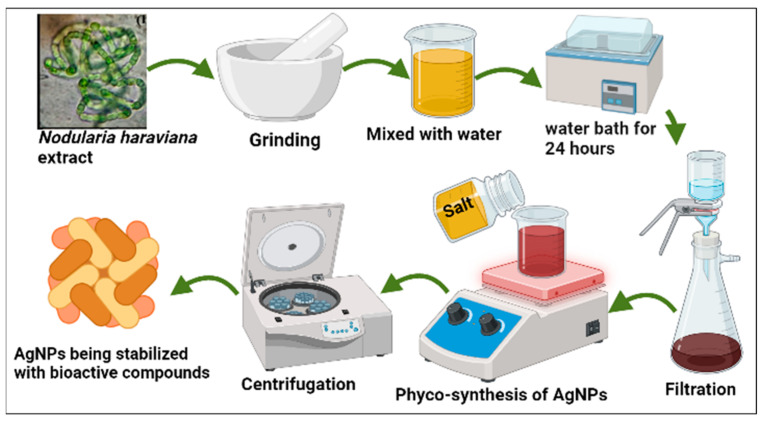
Phyco-synthesis of AgNPs using *Nodularia haraviana.*

**Figure 2 microorganisms-11-02544-f002:**
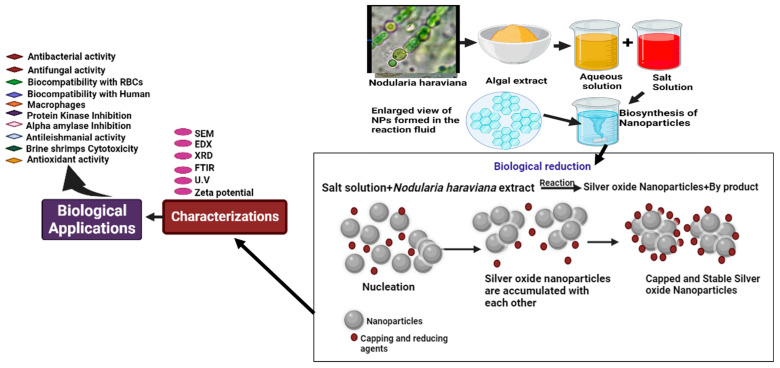
Diagrammatic representation of the synthesis of AgNP nanoparticles via green route.

**Figure 3 microorganisms-11-02544-f003:**
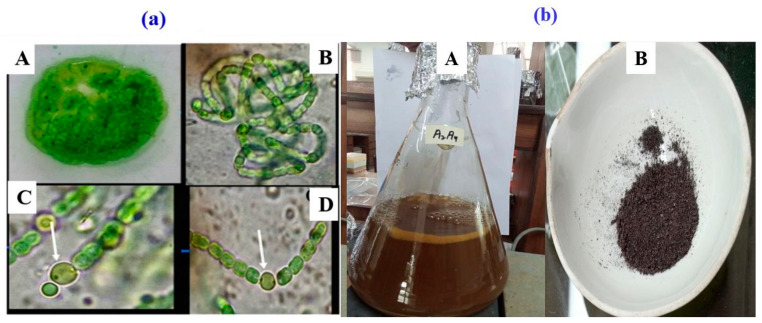
(**a**). (A) Growth of *Nodularia haraviana* on agar plate. (B) Mature filament. (C) Lateral heterocysts (Arrow) (D) Intercalary heterocysts (Arrow) (**b**). (A) Color change indicates the formation of silver nanoparticles. (B) Image of calcined AgNPs.

**Figure 4 microorganisms-11-02544-f004:**
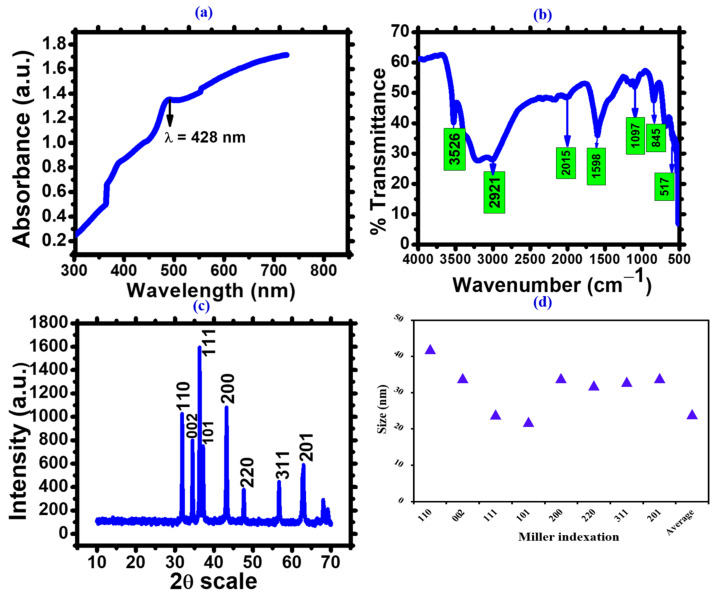
*Nodularia haraviana* mediated silver oxide nanoparticles. (**a**) UV-visible spectrum. (**b**) FTIR spectrum. (**c**) XRD analysis. (**d**) Size calculation via Scherer’s equation.

**Figure 5 microorganisms-11-02544-f005:**
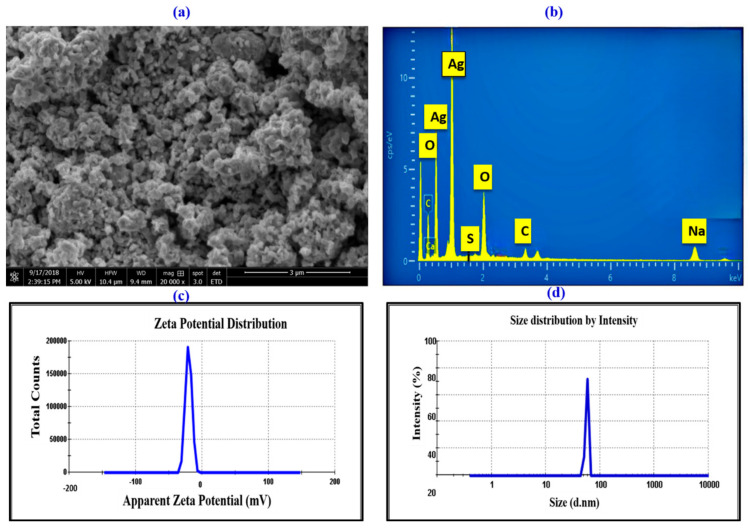
Characterization of phyco-synthesized AgNPs. (**a**) SEM image. (**b**) Elemental composition of AgNPs using EDX. (**c**) ZP measurements. (**d**) Particle size distribution.

**Figure 6 microorganisms-11-02544-f006:**
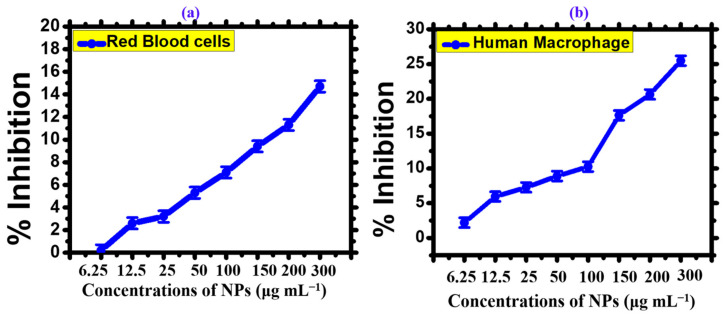
Biocompatibility potential of *Nodularia haraviana* mediated phyco-synthesis of AgNPs. (**a**) Human macrophages (HMs). (**b**) Red blood cells (RBCs).

**Figure 7 microorganisms-11-02544-f007:**
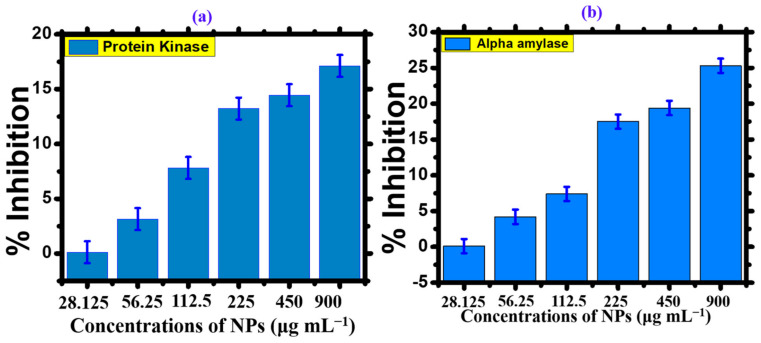
Enzyme inhibition activity of *Nodularia haraviana*-mediated AgNPs. (**a**) Protein kinase inhibition. (**b**) Alpha-amylase inhibition.

**Figure 8 microorganisms-11-02544-f008:**
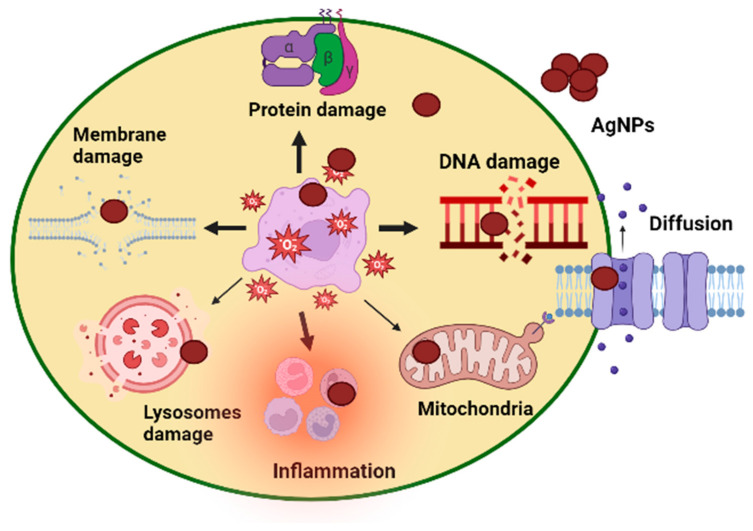
Mechanism of action of AgNP cytotoxicity.

**Figure 9 microorganisms-11-02544-f009:**
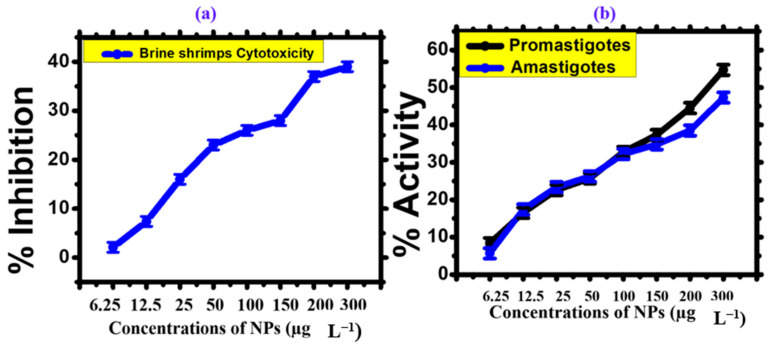
(**a**) Cytotoxicity activity against brine shrimp (BS) (**b**) Cytotoxicity activity against *Leishmania tropica.*

**Figure 10 microorganisms-11-02544-f010:**
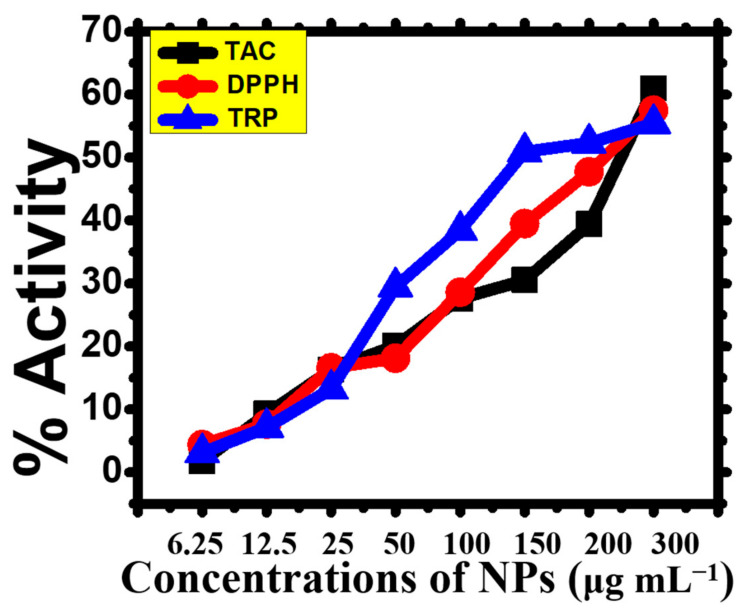
Antioxidant activities (DPPH, TRP, TAC) of Nodularia haraviana-mediated AgNPs.

**Figure 11 microorganisms-11-02544-f011:**
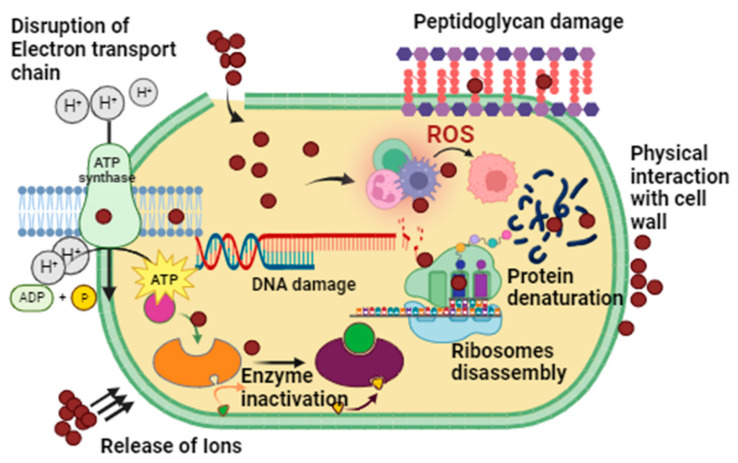
Mechanism of action of AgNPs antimicrobial activity.

**Figure 12 microorganisms-11-02544-f012:**
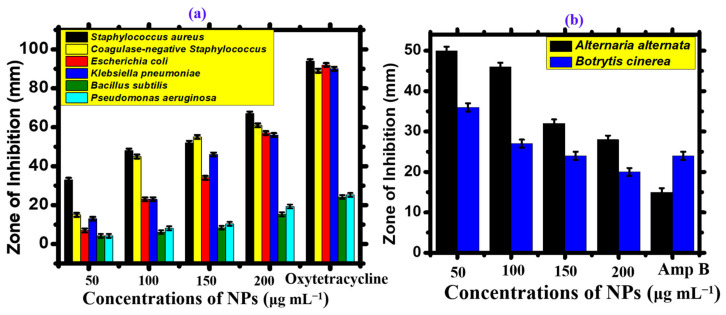
(**a**) Antibacterial activity. (**b**) Antifungal activity.

**Table 1 microorganisms-11-02544-t001:** Analysis of the FTIR spectrum on *Nodularia haraviana*-mediated AgNPs.

Sr. No	Wavenumber cm^−1^	Vibration Type	Appearance	Assignment	Functional Group
1	3526	Stretching	Strong	O–H	Alcohols or Phenols
2	2921	Stretching	Medium	C–H	Alkanes
3	2015	Stretching	Strong	N=C=S	Isothiocyanate
4	1598	Bending	Medium	N-H	Amine
5	1097	Stretching	Strong	C˗O	Alcohol
6	845	Stretching	Strong	C–Cl	Alkyl chloride or Chloroalkane
7	517			AgO	

**Table 2 microorganisms-11-02544-t002:** Conditions for ZP measurements, ZP, PDI, and zeta size distribution.

Conditions for ZP Measurements		Zeta Size Distribution, ZP, and PDI	
Buffer name	PBS	Zeta deviation (mV)	18.1
pH	7.2	Conductivity (mS/cm)	0.132
Zeta runs	12	PDI	1.00
Temperature (°C)	25	Zeta size	179.3.
Count rate (kcps)	70	Zeta average size (d.nm)	632.7
Measurement position (mm)	4.45	Zeta potential	−18 mV
Dispersant RI	1.230	Intercept	0.929
Viscosity (cp)	0.5572	Result quality	Better

**Table 3 microorganisms-11-02544-t003:** Toxicity assessment and calculated biocompatibility test values for AgNPs.

Concentration of AgNPs (µg mL^−1^)	Biocompatibility Test (%)		Toxicity Assessment (%)		
	RBC	Macrophages	*Artemia salina*	*Leishmania tropica*	
				Amastigotes	Promastigotes
6.25	0.2 ± 0.02	2.2 ± 0.02	2.1 ± 0.02	5.7 ± 0.02	8.4 ± 0.02
12.5	2.6 ± 0.02	5.95 ± 0.02	7.4 ± 0.02	17.4 ± 0.02	16.5 ± 0.02
25	3.2 ± 0.02	7.28 ± 0.02	16 ± 0.02	23.4 ± 0.02	22.6 ± 0.02
50	5.3 ± 0.02	8.9 ± 0.02	23 ± 0.02	26.25 ± 0.02	25.7 ± 0.02
100	7.1 ± 0.02	10.26 ± 0.02	26 ± 0.02	32.2 ± 0.02	32.8 ± 0.02
150	9.4 ± 0.02	17.62 ± 0.02	28 ± 0.02	347.8 ± 0.02	37.3 ± 0.02
200	11.3 ± 0.02	20.63 ± 0.02	37 ± 0.02	38.5 ± 0.02	44.5 ± 0.02
300	14.7 ± 0.02	25.48 ± 0.02	39 ± 0.02	47.3 ± 0.02	54.7 ± 0.02
IC50	2124	>395	134.1	38.6	18.71

**Table 4 microorganisms-11-02544-t004:** Calculated antioxidant values for AgNPs.

Concentration of AgNPs (µg mL^−1^)	% Inhibition		
	TAC	DPPH	TRP
6.25	1.7 ± 0.03	4.4 ± 0.03	3.12 ± 0.03
12.5	9.3 ± 0.02	7.6 ± 0.03	7.12 ± 0.03
25	16.3 ± 0.03	16.6 ± 0.03	13.24 ± 0.03
50	20.1 ± 0.03	18.1 ± 0.03	29.5 ± 0.03
100	27.6 ± 0.03	28.6 ± 0.03	38.4 ± 0.03
150	30.5 ± 0.03	39.5 ± 0.03	50.9 ± 0.03
200	39.4 ± 0.03	47.7 ± 0.03	52.3 ± 0.03
300	61 ± 0.03	57.5 ± 0.03	55.4 ± 0.03

**Table 5 microorganisms-11-02544-t005:** Zone of inhibition (ZOI) and minimal inhibitory concentrations (MIC) for different bacterial strains.

Compound	Concentrations (µg mL^−1^)	Zone of Inhibition (mm)					
		Gram-Positive Bacteria			Gram-Negative Bacteria		
AgNPs		*Staphylococcus aureus*	*Coagulase-negative Staphylococcus*	*Bacillus subtilis*	*Escherichia coli*	*Klebsiella pneumonia*	*Pseudomonas aeruginosa*
	50	33 ± 0.02	15 ± 0.02	4.2 ± 0.02	7 ± 0.02	13 ± 0.02	4.2 ± 0.02
	100	48 ± 0.02	45 ± 0.02	6.1 ± 0.02	23 ± 0.02	23 ± 0.02	8.1 ± 0.02
	150	52 ± 0.02	55 ± 0.02	8.4 ± 0.02	34 ± 0.02	46 ± 0.02	10.4 ± 0.02
	200	67 ± 0.02	61 ± 0.02	15.3 ± 0.02	57 ± 0.02	56 ± 0.02	19.3 ± 0.02
Oxytetracycline	10 µg	94 ± 0.02	89 ± 0.02	24.2 ± 0.02	92 ± 0.02	90 ± 0.02	25.2 ± 0.02
Minimal Inhibitory Concentrations (MIC µg mL^−1^)							
AgNPs		MIC	MIC	MIC	MIC	MIC	MIC
		33 ± 0.02	15 ± 0.02	4.2 ± 0.02	7 ± 0.02	13 ± 0.02	4.2 ± 0.02

**Table 6 microorganisms-11-02544-t006:** Zone of inhibition (ZOI) and minimal inhibitory concentrations (MIC) for fungal strains.

Compound	Concentrations (µg mL^−1^)	Zone of Inhibition (mm)	
		Fungal Strains	
AgNPs		*Alternaria alternata*	*Botrytis cinerea*
	50	50 ± 0.02	36 ± 0.02
	100	46 ± 0.02	27 ± 0.02
	150	32 ± 0.02	24 ± 0.02
	200	28 ± 0.02	20 ± 0.02
Ampicillin B	10 µg	15 ± 0.02	24 ± 0.02
Minimal Inhibitory Concentrations (MIC µg mL^−1^)			
AgNPs		MIC	MIC
		50 ± 0.02	36 ± 0.02

**Table 7 microorganisms-11-02544-t007:** Algal mediated phyco-synthesis of AgNPs and their biomedical applications.

Algal Strains	Method	Size (nm)	Morphology	Applications	References
*Spirulina platensis*	Green synthesis	2.23–14.68 nm	spherical	Anticancer activity	[61]
*Nostoc carneum*	Green synthesis	4–22 nm	quasi-spherical	antibacterial, anticoagulative photocatalytic degradation	[11]
*Dunaliella salina*	Green synthesis	35 nm	spherical	Antibacterial activity	[62]
*Microcystis* sp.	Green synthesis	40–130 nm	cubic, spherical, and rod	Antimicrobial activity	[63]
*Nodularia haraviana*	Green synthesis	24.1 nm	spherical	Antimicrobial, cytotoxicity biocompatibility with human RBC and macrophages, enzyme inhibition potential and their antioxidant capacities	Current study

## Data Availability

The data supporting this study’s findings are available from the corresponding author upon reasonable request.

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
