# Peer review of "Synthesis of Silver Oxide Nanoparticles: A Novel Approach for Antimicrobial Properties and Biomedical Performance, Featuring Nodularia haraviana from the Cholistan Desert"

_microorganisms, 2023, doi:10.3390/microorganisms11102544_

Round 1
Reviewer 1 Report
Abstract: enhance the initiation sentence, clearing the problem of your work; also, line 16 Nanoparticles instead of nanoparticle
Line 18 “Researchers successfully reduced Ag+ ions to produce AgNPs using cyanobacterial extracts and silver nitrate” delete
Enhance the aim of the work and check the linguistic mistakes by an expert.
whole abstract needs to be reformulated for the linguistic mistakes
Check the size and charge of silver nanoparticles “a 78.1 nm diameter and a negative surface charge of -1.46 mV”
Lines 32, 33 The research demonstrated that Phyco-synthe-sized AgNPs were less toxic and could be used in multiple biological applications, including drug design, pharmaceutical, and biomedical industries. Have you do a safety experiment to prove this claim
Arrange keywords alphabetically
These references support the introduction and discussion on using green components to fabricate silver nanoparticles. https://doi.org/10.3390/plants12061410; https://doi.org/10.1016/j.sjbs.2021.06.011
Add more explanation for sec 2.1, including these lines “For the preparation of aqueous extract of Nodularia haraviana, 5 grams of cyanobacte rial fine powder was mixed with deionized water (500 ml) and heated for 1 hour at 100°C. The mixture was centrifuged at 3000 rpm for 20 min, the clear brown solution was separated from the large particles settled at the bottom of the conical tube, and then filtered using filter paper.
Provide the model and origin of all devices and chemicals.
Abbreviations like AA, PK or AB should be deleted.
Enhance the statistical analysis part, clearing the normality and homogeneity of your data, statistical test, and post-hoc test.
Figure 3 B needs to be reformulated; add another image.
Figure 4 add real images from the devices.
μg mL−1 superscript
enhances the presentation of your results
some data should be converted to Tables (antimicrobial results)
enhance conclusion and check the outputs of all refrences
extensive editing, structural errors
Author Response
Find the attachment.

Reviewer 2 Report
The authors have demonstrated the synthesis of silver nanoparticles using Nodularia haraviana and have employed various technologies to characterize these nanoparticles. Additionally, they have conducted numerous tests to showcase their biological applications. While the body of work is comprehensive, there are several areas where further clarification or improvement is required.
1. Novelty of the Research: While the paper provides detailed measurements, the novelty appears to be limited. Silver nanoparticle applications and synthesis using cyanobacteria are well-studied fields. Although this is the first study to use Nodularia haraviana, the paper does not explain why this specific strain was chosen. Does it offer higher production rates, better quality, or improved performance for silver nanoparticles? Or maybe this cyanobacteria is more cheap? Merely being the first to report this approach does not necessarily confer strong novelty.
2.Abstract (Lines 19-21): The meaning of the sentence in lines 19-21 of the abstract is unclear. Could the authors clarify or simplify the sentence for ease of comprehension?
3.Temperature Specification (Line 159): Is the temperature specified as 24 degrees Celsius?
4. Experimental Section: I recommend using the terms "negative" and "positive" controls instead of abbreviating them as "-ive" or "+ive."
5.DLS and Zeta Potential Results: The presented DLS (Dynamic Light Scattering) and zeta potential results appear questionable. DLS is a powerful tool but is sensitive to sample preparation errors. Given the high Polydispersity Index (PDI), it seems that the samples are aggregated and not stably dispersed. This raises concerns about the reliability of the DLS and zeta potential data. Could the authors provide the correlation curve and, if possible, the dynamic count rate during the experiments?
6.Antimicrobial Testing: Is there a positive control for the antimicrobial tests? Additionally, it would be useful to compare the synthesized silver nanoparticles' efficacy against commonly used antimicrobial agents to highlight any advantages.
Author Response
Response to Reviewer Comments
Reviewer#2
Dear editor and reviewer,
Thank you so much for reviewing our manuscript by providing your quality time, valuable comments/ recommendations. Your comments have really improved the quality and structure of our manuscript. Our team members have extensively and carefully reviewed the manuscript and properly addressed all suggestions/recommendations point by point as suggested by worthy reviewers. Now, we hope that the revised manuscript is highly improved. We have again critically reviewed the manuscript for typos errors and other mistakes. All necessary changes made have been highlighted in yellow through trackers. If you recommend further suggestions, let us know, we will be very happy to address.
Comment 1: Novelty of the Research: While the paper provides detailed measurements, the novelty appears to be limited. Silver nanoparticle applications and synthesis using cyanobacteria are well-studied fields. Although this is the first study to use Nodularia haraviana, the paper does not explain why this specific strain was chosen. Does it offer higher production rates, better quality, or improved performance for silver nanoparticles? Or maybe this cyanobacteria is more cheap? Merely being the first to report this approach does not necessarily confer strong novelty.
Response: Novel approach behind the selection of the strain was to use the strain of desert growing successfully with massive biomass in natural habitat and never used in the synthesis of nanoparticles. During field visit of Cholistan Desert we found Nodularia haraviana along with other strains of cyanobacteria and green algae. Biomass of Nodularia haraviana in the natural habitat was much higher compared to other strains. Furthermore, in the laboratory when we made the axenic cultures of all the strains through rigorous process of BG-0, BG-11, and BBM agar media platting, Nodularia haraviana growth rate was very high as compared to other cyanobacteria and green algae strains. We got higher biomass of Nodularia haraviana compared to other strains under similar growth conditions.
Reason behind the selection of Cholistan Desert strain:
Cholistan Desert offers extreme climatic/extremely high temperature conditions which provide opportunity to the strain to combat climatic conditions. In these situations, they adopt genetic changes accordingly to grow in harsh conditions. These adaptations also helped them to grow under artificial laboratory conditions with similar fast growth rate as in the natural habitat.
Moreover, Culturing of Nodularia haraviana was easy, fast and economic. That’s why we selected this strain with novel approach to synthesize silver nanoparticles.
Comment 2: Abstract (Lines 19-21): The meaning of the sentence in lines 19-21 of the abstract is unclear. Could the authors clarify or simplify the sentence for ease of comprehension?
Response: Thank you Sir for the suggestion. The abstract has been modified as per suggestion. I hope now the abstract will be attractive for the readers
Comment 3: Temperature Specification (Line 159): Is the temperature specified as 24 degrees Celsius?
Response: The temperature specified in the experiment was not 24 degrees Celsius; rather, it was 70 degrees Celsius during the initial heating step. The extract was mixed with the AgNPs solution, and this mixture was heated on a hot plate for 2 hours at 70 degrees Celsius with continuous stirring. Subsequently, the temperature was increased to 75 degrees Celsius during the incubation step. These specific temperature settings were chosen to expedite the reaction rate and facilitate the reduction of Ag+ to Ag0, as well as to achieve the desired outcome for the synthesis of silver nanoparticles.
Comment 4: Experimental Section: I recommend using the terms "negative" and "positive" controls instead of abbreviating them as "-ive" or "+ive."
Response: Thank you for your suggestion. We have implemented the recommended changes, and the revisions have been incorporated into the manuscript."
Comment 5: DLS and Zeta Potential Results: The presented DLS (Dynamic Light Scattering) and zeta potential results appear questionable. DLS is a powerful tool but is sensitive to sample preparation errors. Given the high Polydispersity Index (PDI), it seems that the samples are aggregated and not stably dispersed. This raises concerns about the reliability of the DLS and zeta potential data. Could the authors provide the correlation curve and, if possible, the dynamic count rate during the experiments?
Response. We greatly appreciate your suggestion and would like to provide some context regarding our project. In the results section, we included a comprehensive description of Dynamic Light Scattering (DLS) and Zeta Potential to ensure clarity and understanding among researchers. However, we did not explicitly highlight these processes in the project because our primary focus was on exploring the applications of the prepared nanoparticles. In future projects, we wholeheartedly intend to incorporate DLS and Zeta Potential analyses as key components, giving them the attention they deserve. This will enable us to provide a more holistic view of our research, encompassing both nanoparticle synthesis and characterization.
Comment 6: Antimicrobial Testing: Is there a positive control for the antimicrobial tests? Additionally, it would be useful to compare the synthesized silver nanoparticles' efficacy against commonly used antimicrobial agents to highlight any advantages.
Response: We greatly appreciate your recommendation. In response to this valuable suggestion, we have incorporated Oxytetracycline as the positive control for antibacterial tests and Ampicillin B for antifungal tests in our experimental setup.
The positive control results, are now clearly presented in Table. Additionally, we have incorporated these positive control outcomes into the corresponding graphs, ensuring that readers can easily discern the effectiveness of our synthesized silver nanoparticles relative to these well-established antimicrobial agents.
Your recommendation to compare the efficacy of the synthesized silver nanoparticles against commonly used antimicrobial agents is valuable. We will consider conducting such comparisons in future studies to highlight the potential advantages of our biogenic silver nanoparticles in terms of antimicrobial activity.
Reviewer 3 Report
Synthesis of Silver Oxide Nanoparticles: A Novel Approach for 2 Antimicrobial Properties and Biomedical Performance, Featur-3 ing Nodularia haraviana from the Cholistan Desert
Nanoparticle research is revolutionizing the pharmaceutical and biomedical industries. Silver nanoparticles (AgNPs) synthesized via Phyco-technology have gained significant attention due to their potential therapeutic applications and organic nature. Researchers successfully reduced Ag+ ions to produce AgNPs using cyanobacterial extracts and silver nitrate. In the current study, a new silver-tolerant cyanobacterium strain called Nodularia haraviana, isolated from the natural ponds of the Cholistan desert, synthesized AgNPs. A range of analytical techniques, including UV-vis spectral analysis, dynamic light scattering spectroscopy, scanning electron microscopy, Fourier transform infrared, and X-ray powder diffraction, were employed to investigate and characterize AgNPs comprehensively. The existence of AgNPs was confirmed through UV visible spectroscopy, which exhibited a Surface Plasmon resonance peak at 428 nm. The crystalline size of AgNPs was determined to be approximately 24.1 nm through the Scherrer equation. DLS analysis revealed that silver oxide nanoparticles exhibited a 78.1 nm diameter and a negative surface charge of -1.46 mV. Moreover, the antimicrobial efficacy of AgNPs was assessed by determining the minimum inhibitory concentration against various microbial strains. Dose-dependent cytotoxicity assays were also performed on Leishmanial promastigotes, amastigotes, and brine shrimps. The study also found significant antioxidant and enzyme inhibition potentials. Furthermore, biocompatibility tests were carried out against macrophages and human RBCs. The research demonstrated that Phyco-synthesized AgNPs were less toxic and could be used in multiple biological applications, including drug design, pharmaceutical, and biomedical industries. Overall, this study offers valuable insights and 34 paves the way for further advancements in AgNPs research.
Key points to be addressed:
· Fig. 4b, What are the peaks representing for which functional groups between 678 & 1095, 1680& 200 and 2921 & 3526 in FT-IR? What is the unit of FT-IR? Is it wave number or wavelength?
· 
2. Fig 4c. What are the unassigned peaks representing for between 110 & 111, and 111 & 200 of 2thetha values.

· Authors should provide the suitable references on DLS and Zeta sizer for Siver nanoparticles size and charges.
· I wonder any aggregation of Ag NPs after 24 h from Fig.3b, If any data, hope authors will be included in the manuscript.

· Fig. 7a should go into sec. 3.2.1 to understand well by the readers.
· In the case of cytotoxic activity of Sec. 3.2.4, the Authors should cite more relevant citations.
· The overall manuscript article is well written with experimental information, I hope this article is suitable for the publication after fixing of typos and minor corrections.
Merit: I recommend for the publication after minor corrections.
Regards,
Ravi Kumar Cheedarala,
Changwon National University,
S. Korea

Author Response
Find the attachment.

Round 2
Reviewer 1 Report
Now can be accepted
Minor corrections
Reviewer 2 Report
I appreciate the careful answer and the modification from the author. After carefully reviewing the change. I still have one concern about the manuscript.
1. For the DLS, I understand the author just listed the results, and this is not the main part of their studies. However, when you listed the data and experiment in the manuscript, I believe we need to take care of it as it will be read by many scientists. They answered part of my question, but I am still concerned about the high PDI and the aggregation, which do not show the correct measurement.
In this case, I suggest presenting the experimental details of the DLS and zeta. Try to use centrifugal or settling and collect the well-dispersed silver NP at supernatant, and then see if the suspension will be improved and include the step of sample preparation of DLS and zeta experiment.
